# CRITICAL LEARNING PERIODS EMERGE EVEN IN DEEP LINEAR NETWORKS

**Michael Kleinman**[1]  **Alessandro Achille**[2]  **Stefano Soatto**[3]
[1]Stanford University    [2]Caltech    [3]UCLA
mkleinman@stanford.edu aachille@caltech.edu soatto@ucla.edu

## ABSTRACT

Critical learning periods are periods early in development where temporary sensory deficits can have a permanent effect on behavior and learned representations. Despite the radical differences between biological and artificial networks, critical learning periods have been empirically observed in both systems. This suggests that critical periods may be fundamental to learning and not an accident of biology. Yet, why exactly critical periods emerge in deep networks is still an open question, and in particular it is unclear whether the critical periods observed in both systems depend on particular architectural or optimization details. To isolate the key underlying factors, we focus on deep linear network models, and show that, surprisingly, such networks also display much of the behavior seen in biology and artificial networks, while being amenable to analytical treatment. We show that critical periods depend on the depth of the model and structure of the data distribution. We also show analytically and in simulations that the learning of features is tied to competition between sources. Finally, we extend our analysis to multi-task learning to show that pre-training on certain tasks can damage the transfer performance on new tasks, and show how this depends on the relationship between tasks and the duration of the pre-training stage. To the best of our knowledge, our work provides the first analytically tractable model that sheds light into why critical learning periods emerge in biological and artificial networks[1].

## 1 INTRODUCTION

Critical learning periods are time periods early in development where temporary sensory deficits can permanently damage the outcome of learning. In biology, critical periods have been studied systematically since Hubel and Wiesel analyzed visual development in kittens (Wiesel & Hubel, 1963). Critical learning periods have since been shown to exist across different learning skills (vision and language), different species (kittens, dogs, and humans) and different sensory modalities (visual and auditory) (Kandel et al., 2013).

The most widely accepted explanation for the existence of critical learning period phenomena has to do with the characteristics of biological hardware: As the brain ages, biochemical processes decrease neural plasticity, making it increasingly difficult to form new synaptic connections and to learn new skills (Hensch, 2004). Surprisingly, phenomena analogous to critical learning periods have been empirically observed for artificial deep neural networks (DNNs) (Achille et al., 2019; Kleinman et al., 2023), suggesting critical learning periods may be a more general feature of agents learning, and not caused directly by biologically ascribed factors like changing plasticity or inhibition.

Since the biochemical explanation does not hold for artificial systems, it is unclear what may cause them: One possibility is that critical periods in artificial DNNs could be due particularities of the optimization (e.g., an annealing learning rate); alternatively it could arise from defects in the artificial implementation and training (e.g., ReLU units becoming frozen or gradients vanishing). If that were the case, it would be difficult to argue the connection with biological systems, which would not have these issues.

In this paper, we establish for the first time that critical periods can exist in a minimal analytical model of deep networks: deep linear networks (which do not suffer from any of the above). Deep linear

---

[1]Code available at: https://github.com/mjkleinman/CriticalPeriodDeepLinearNets

networks (Saxe et al., 2013) are deep networks without non-linearities between layers. We consider two related, but distinct cases, of overparametrized deep linear networks that capture distinct critical periods phenomena of interest. In Sect. 3, we consider a multipathway deep linear network (Shi et al., 2022) to study how competition between pathways affect how features get learned. In Sect. 4 we consider the setting of matrix completion using a deep linear network parameterization (Gunasekar et al., 2017; Arora et al., 2019) to study how generalization is impacted by initially training using a different data distribution. We show that both settings are analytically tractable, with differential equations that characterize learning dynamics underlying such phenomena.

The multi-pathway model parameterization allows us to simulate different competing ways to explain the data. Biological systems often exhibit critical periods that depend on a complex interaction between sensors. For example, in a classical experiment, Hubel and Wiesel showed that occluding one eye early in development leads to permanent loss of vision in that eye (Wiesel & Hubel, 1963). However, if a part of the retina of the uncovered eye is damaged, the other eye still learns to exploit the limited information (Guillery, 1972). In our model, we are able to reproduce similar critical periods and competition/inhibition between sensors as observed in biological models. We show analytically and in simulation that the learning of features is tied to competition between sources. The learning of the singular values occurs in a "race", similar to a winner-take-all structure, with both pathways competing to produce the output and the competition becoming more pronounced as the depth of the network is increased.

The matrix completion setting allows a natural notion of generalization, as well as a well defined notion of tasks, their complexity, and their relationships. We show that pre-training on certain tasks can damage the transfer performance on new tasks. This occurs if brittle features from the initial task can sufficiently explain the data on the final task, and this effect becomes more pronounced in deeper networks. Such complex interaction is, again, not a function of the complex architecture, or the complex optimization or implementation, but is manifest even in tractable deep linear networks.

Overall, our analysis shows that critical periods in deep networks depend primarily on two main factors: the depth of the model and the structure of the data distribution, as opposed to details of the architecture and optimization problem. This level of abstraction allows us to establish a strong correspondence with biological systems. From a neuroscience perspective, our analysis provides an alternative explanation of critical periods that does not hinge on biochemical changes in plasticity, but is rather fundamental to learning, as observed in radically different embodiments. Our analysis and the empirical evidence we uncover in the tractable deep linear network setting may also provide tools for deep learning practitioners to better understand transfer learning and multimodal learning.

## 2 RELATED WORK

**Critical learning periods in humans and other animals.** Critical learning periods are time windows early in development where temporary sensory deficits can permanently impair behavior and alter learned representations (Wiesel, 1982; Kandel et al., 2013; Knudsen, 2004). In biology, critical periods have been studied systematically since Hubel and Wiesel analyzed visual development in kittens (Wiesel & Hubel, 1963; Hubel & Wiesel, 1970) and have since been shown to exist across different learning skills (vision and language), different species (kittens, dogs, and humans) and many different sensory modalities (visual, auditory, and motor) (Kral, 2013; Kandel et al., 2013). While critical learning periods are typically studied by altering the sensory information an animal is exposed to early during development, these learning periods also provides a system with the ability to flexibly adapt to their particular environment (Kandel et al., 2013).

**Critical periods in artificial networks.** Achille et al. (2019) found that deep neural networks exposed to blurred images early during training exhibited phenomena analogous to animals exposed to a similar deficit. More recently, Kleinman et al. (2023) found that DNNs also had critical learning periods for multisensory integration, with deficits early in training affecting both the learned representations and behaviour. Also related, Golatkar et al. (2019) found that regularization applied during this early period of training had the most significant effect on generalization performance.

**Learning dynamics in deep linear networks.** A deep linear network is a deep neural network, with identity for the activation function. Although the input-output map is linear, the learning dynamics are nonlinear (Saxe et al., 2013). Moreover, it is possible to obtain exact learning dynamics based on how the network will learn task structure (Saxe et al., 2013), and such networks have provided insight

into semantic development in humans (Saxe et al., 2019). Recently, such models have been extended to the multi-pathway setting, where it was shown that deeper networks are increasingly likely to learn features of either pathway (but not share on both) (Shi et al., 2022). It is also possible to add gating to deep linear networks to alter the flow of information, and this can allow nonlinear computation. (Saxe et al., 2022). Here, we incorporate both a gating and multi-pathway extension to study critical learning periods in deep linear networks.

**Matrix completion, deep matrix factorization, and the implicit regularization of SGD.** Matrix completion is a general problem of imputing missing values in a matrix, given some observed entries. This setting is common, and often seen in recommendation systems where only a fraction of recommendation (or ratings) are known. If the ground-truth matrix is low-rank (often the case for real-world data), missing values can recovered given sufficient number of observed entries by minimizing over matrices that match the observed entries and have minimum nuclear norm (Candes & Recht, 2012). Gunasekar et al. (2017) took a different perspective on the matrix completion problem, and empirically found that parametrizing the target matrix using two layer linear neural network, that is $W = W_2 W_1$, and optimizing over the weights of this factorization using gradient descent with small initialization lead to matrices that had minimum nuclear norm, suggesting that the implicit bias of gradient descent was to find low nuclear norm solutions. Afterwards, Arora et al. (2019) refined the results, finding that gradient descent starting from a small initialization is implicitly minimizing the rank of the matrix, as opposed to the nuclear norm, and this effect is increasingly more pronounced in deeper networks. We use the matrix completion setup to study how generalization is affected if a network did not start from small initializations, but rather was initially trained on another task.

## 3 IMPACT OF DEPTH AND TEMPORARY DEPRIVATION ON FEATURE LEARNING IN MULTI-PATH MODEL

Inputs are rarely processed by a single processing stream. For example, humans and other animals have two eyes that process visual information coming from a scene. In a series of systematic biological experiments, researchers discovered that kittens with a single eye occluded early during development were more affected than kittens with both eyes occluded during the same period. This suggests complex dynamics are mediating learning from multiple sensory modalities. Here, we analyze how features get learned in a minimal multi-patwhway model, stripped of nonlinearities. In particular, we study what happens if a pathway becomes temporarily blocked, such as from suturing a eye. Surprisingly, as we will see, this minimal and analytically tractable model captures much of the learning dynamics seen in biological experiments.

### 3.1 LINEAR MULTI-PATHWAY FRAMEWORK

We consider a multipath linear network (Shi et al., 2022) where the output $\mathbf{y}$ is produced by propagating an input $\mathbf{x}$ through multiple pathways $\mathcal{P} = \{a, b\}$ as follows:

$$\mathbf{y} = \mathbf{W}_a^{D_a} \cdots \mathbf{W}_a^2 \mathbf{W}_a^1 \mathbf{x} \ + \ \mathbf{W}_b^{D_b} \cdots \mathbf{W}_b^2 \mathbf{W}_b^1 \mathbf{x} \tag{1}$$

$$= \Big( \sum_{p \in \mathcal{P}} \mathbf{W}_p^{D_p} \cdots \mathbf{W}_p^2 \mathbf{W}_p^1 \Big) \mathbf{x} = \mathbf{\Omega} \mathbf{x}, \tag{2}$$

where $\mathbf{W}_p^d$ denotes the $d^{\text{th}}$ weight matrix along pathway $p$. We will focus on the case where $|\mathcal{P}| = 2$ pathways. As in Shi et al. (2022) we have defined $\mathbf{\Omega}_p \equiv \prod_{d=1}^{D_p} \mathbf{W}_p^d$, so

$$\mathbf{\Omega} = \mathbf{\Omega}_a + \mathbf{\Omega}_b = \sum_{p \in \mathcal{P}} \mathbf{\Omega}_p \equiv \sum_{p \in \mathcal{P}} \prod_{d=1}^{D_p} \mathbf{W}_p^d. \tag{3}$$

The input $\mathbf{x}$ thus gets propagated through multiple deep pathways of depth $D_p$, with each pathway consisting of a series of linear transformations $\prod_{d=1}^{D_p} \mathbf{W}_p^d$. We consider the case of minimizing the squared error loss

$$L = \frac{1}{2} \sum_{i=1}^{N} ||\mathbf{y}^{(i)} - \mathbf{\Omega} \mathbf{x}^{(i)}||^2 \tag{4}$$

with a training set of i.i.d. samples $\mathcal{D} = \{(\mathbf{x}^{(i)}, \mathbf{y}^{(i)})\}_{i=1}^{N}$. To simplify even further, we assume that the inputs have been whitened, so that the input correlation matrix $\mathbf{\Sigma}^x = \frac{1}{N} \sum_i \mathbf{x}^{(i)} \mathbf{x}^{(i)T} = \mathbf{I}$.

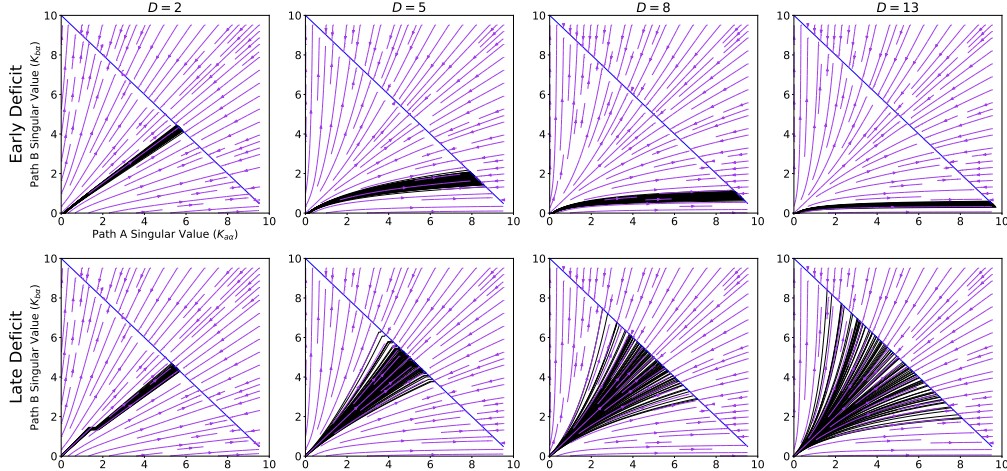

Figure 1: **Deeper networks are more affected by a temporary early deficit.** Phase portrait of pathway specific singular value $K_{a\alpha}$ and $K_{b\alpha}$ where $K_{a\alpha} + K_{b\alpha} = \sigma_\alpha$, where $\sigma_\alpha = 10$ is the singular value and denoted by the diagonal blue line in the plots. Black traces indicate a simulation for a particular initial condition (100 shown). Flow fields (purple arrows) are shown for systems without a deficit. **(Top row)** An early deficit to one pathway ("B") leads to the other pathway learning more of the feature, and this effect becomes increasingly more pronounced in deeper networks (shown for depths ranging from 2 to 13). **(Bottom row)** In contrast, late deficits have a negligible effect on the final solution, where neither pathway dominates (on average) how a feature is learned.

Let $\Sigma^{yx} = \frac{1}{N} \sum_{i=1}^{N} \mathbf{y}^{(i)} \mathbf{x}^{(i)T}$ be the cross-correlation matrix between the inputs $\mathbf{x}$ and the target vector $\mathbf{y}$ and let $\Sigma^{yx} = \mathbf{U}\mathbf{S}\mathbf{V}^T$ be its singular-value decomposition (SVD). The loss will be minimized when $\Sigma^{yx} = \Omega$, and hence when $\bar{\Omega} \equiv \mathbf{U}^T \Omega \mathbf{V} = \mathbf{S}$, or the network has learned the task-appropriate singular values. We define $\mathbf{K}_a = \mathbf{U}^T \Omega_a \mathbf{V}$ as the pathway specific contribution to the singular values for path $a$ (the contribution of both pathways $\mathbf{K}_a + \mathbf{K}_b$ sum to $\mathbf{S}$ at convergence).

Using the continuous time limit of the SGD update equation leads to the following differential equation for pathway $a$ (and analogously for pathway $b$):

$$\tau \frac{d}{dt}\mathbf{W}_a^d = \Big( \prod_{i=d+1}^{D_a} \mathbf{W}_a^i \Big)^T (\Sigma^{yx} - \Omega\Sigma^x) \Big( \prod_{i=1}^{d-1} \mathbf{W}_a^i \Big)^T. \tag{5}$$

Assuming $\mathbf{W}_a^d = \mathbf{R}_a^{d+1} \bar{\mathbf{W}}_a^d \mathbf{R}_a^{d^T}$ where $\mathbf{R}$ is an orthogonal matrix and $\mathbf{R}_a^1 = \mathbf{V}$ and $\mathbf{R}_a^{D_a+1} = \mathbf{U}$ we get

$$\tau \frac{d}{dt}\bar{\mathbf{W}}_a^d = \Big( \prod_{i=d+1}^{D_a} \bar{\mathbf{W}}_a^i \Big)^T (\mathbf{S} - \bar{\Omega}) \Big( \prod_{i=1}^{d-1} \bar{\mathbf{W}}_a^i \Big)^T, \tag{6}$$

and note that if all weight matrices $\bar{\mathbf{W}}_a^i$ are diagonal at initialization, then since $\mathbf{S}, \bar{\Omega}$ are diagonal, we can arrive at a system of scalar differential equations, where for each singular value $S_\alpha$

$$\tau \frac{d}{dt}q_{a\alpha} = q_{a\alpha}^{D_a-2} p_{a\alpha} \left[ S_\alpha - \bar{\Omega}_\alpha \right]$$
$$\tau \frac{d}{dt}p_{a\alpha} = q_{a\alpha}^{D_a-1} \left[ S_\alpha - \bar{\Omega}_\alpha \right] \tag{7}$$

where $q_{a\alpha}$ reflects the scale of diagonal entries of the intermediary matrices ($d < D_a$) while $p_{a\alpha}$ reflects the scale of diagonal entries of the final matrix ($d = D_a$). Note further that the ratio of Eq. 7 will be constant such that:

$$\frac{dq_{a\alpha}}{dp_{a\alpha}} = \frac{p_{a\alpha}}{q_{a\alpha}} \tag{8}$$

so $q^2 - p^2 = q(0)^2 - p(0)^2$ and this difference will stay constant during training.

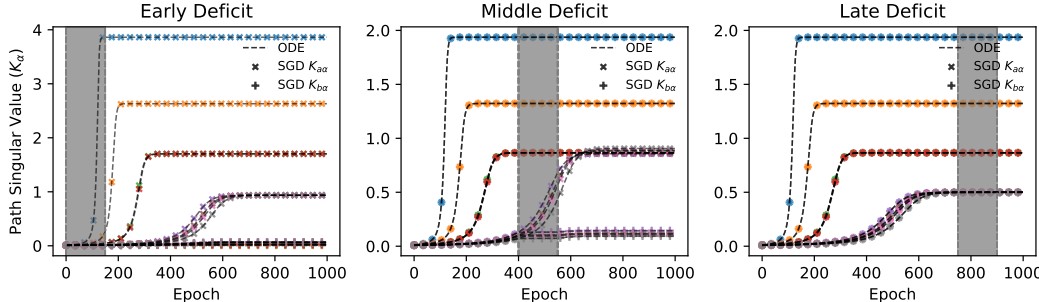

Figure 2: **Early deficits affect learned representations in multi-pathway model, while late deficits do not. (Left)** Early gating deficit (denoted by gray period; epoch 0 to 150) to pathway B leads to all features being learned in pathway A. When training with SGD, we indicate the singular values for pathway A with crosses ('X'), and pathway B with plus signs ('+'), with the different colors identifying different singular values. In dashed lines, we show the results of integrating the ODE (Eq. 7) for each singular value for both pathways. We observe a match between the differential equation and the learning dynamics obtained with SGD training. We observe sigmoidal learning trajectories of the singular values, with larger singular values learned earlier in training. **(Middle)** Deficits applied in the middle of training only affects previously unlearned features (bottom four singular modes), whereas other singular values are learned equally in both pathways (crosses and plus signs overlapping for blue, orange, red, green singular modes). **(Right)** Late deficits (epoch 750 to 900) has a negligible effect on how features are learned, and results in features being learned equally in both pathways (crosses and plus signs overlapping for all singular values).

## 3.2 LEARNING DYNAMICS IN REDUCED SCALAR DIFFERENT EQUATION HIGHLIGHT EFFECT OF COMPETITION

To better understand how competition between processing pathways mediate learning, we introduce deprivation deficits during different windows of training where we temporarily prevent learning (parameter updates) in a deprived pathway. This deficit can also be interpreted as blocking input information from being processed by the deprived pathway (see Appendix A.1 for details), and so we refer to the deficit as a *gating* deficit.

Using the multi-pathway setup described above, we integrate the differential equation of Eq. 7 and plot the corresponding flow fields in Fig. 1. To integrate the differential equation, we use a step size of $\lambda \equiv \frac{1}{N\tau} = 0.001$ in discrete time for 1000 epochs to ensure convergence. To better understand how temporary deficits impact how features get learned, we applied a gating deficit early (first 15 epochs) and late (epoch 100 to 115) in training to one pathway ("B"). In this way, parameter values along the deprived pathway remain constant during the deficit.

We initialize each pathway independently with $p(0) \sim \mathcal{N}(0, 0.01^2)$ and $q^2(0) - p^2(0) = 1$ in Fig. 1, and also observe similar trends if $p(0) = \epsilon$ and $q(0) = \epsilon$ corresponding to a small initialization in Appendix Fig. 7, even though the deficit is only applied during the initial phase of a sigmoidal learning trajectory. We find that competition becomes more pronounced at greater depths; in this manner temporary deficits (of a fixed number of epochs) alters the learning of features more in deeper networks (Fig. 1). Further, only gating deficits early during training affect how features get learned; gating deficits applied late in training do not alter the features that get learned by each pathway (Fig. 1, bottom row).

## 3.3 DEEP MULTI-PATHWAY LINEAR NEURAL NETWORK SIMULATIONS

We next verified that the results from integrating the differential equation corresponded to real learning motifs by training a deep multi-pathway linear network on a previously studied hierarchical task (Shi et al., 2022) (See Appendix A.1). We trained a depth $D_a = D_b = 4$ network, with 100 units per layer of each pathway using SGD with a constant learning rate of 0.01 using the squared error loss (Eq. 4). Again, we find that early deficits affect learned representations, while late deficits do not (Fig. 2). In particular, an early gating deficit (epoch 0 to 150; left) to pathway "B" leads to all features being learned in the other pathway ("A").

This can be better understood by examining the learned singular values for both pathways, which are only learned by the normal pathway (denoted by crosses for SGD simualtions) when the deficit is

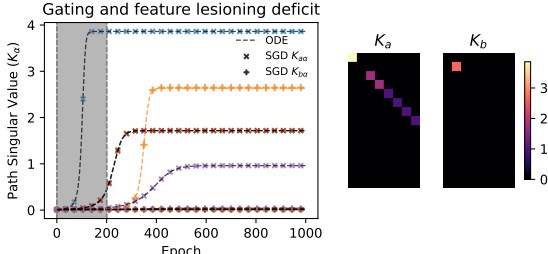

Figure 3: **We can reproduce and better understand classical experiments showing competition between eyes using our multi-path framework.** Applying a deprivation deficit to one pathway (analogous to eye suture; in this case pathway to "B") for first 200 epochs, while permanently lesioning a feature in the other pathway (second singular mode of pathway "A") leads to the initially sutured pathway only learning the corresponding lesioned feature (orange plus signs, and corresponding singular value in $\mathbf{K}_b$). Integration of the ODE (Eq. 7 for each singular value matches simulations with SGD training before, during, and after the deficit (shown in dashed lines). Singular value dimensions 1 to 5 shown for improved readability. This highlights how the pathways compete to learn the input features, and shows that a deprivation deficit to one pathway will result in the other pathway "winning the competion" and learning the corresponding feature(s). This experiment recapitulates classical experiments by Guillery that lesioned a local region of the normal eye during a monocular deprivation experiment, which was intended for studying how competition between eyes/pathways affect how visual features get learned (Guillery, 1972).

applied early during training. This early deficit led to the normal pathway "winning the competition" to learn the singular values and explain the output. In contrast, deficits during the middle of training (Fig. 2, center) only affects singular values that were not previously learned (bottom singular modes). The late deficit (Fig. 2, right) does not affect the previously learned features. The learning of the singular values has a particular sigmoidal learning trajectories, where the network learns the singular value in order of their magnitude (higher singular values learned earlier in training), as we will discuss more in the next section, and in line with prior work (Saxe et al., 2013; Arora et al., 2019). We also observe a match between the differential equation and the learning dynamics obtained with SGD training and integrating the ODE of Eq. 7 for both pathways before, during, and after the deficit period (Fig. 2). We observe similar learning dynamics and effect of a temporary gating deficit in nonlinear networks with a Tanh or ReLU activation function (Fig. 11 and Fig. 12 respectively).

To better understand how competition and inhibition between sensors affects how features are learned, we applied a gating deficit to one pathway (analogous to eye suture), while permanently lesioning a feature in the other pathway (a singular mode). Without the lesioning, the normal pathway learned all the features (Fig. 2), and in this case it resulted in the initially deprived pathway only learning the corresponding lesioned feature (here, the second singular mode, orange plus signs in Fig. 3). This experiment recapitulates classical experiments by Guillery in which a local lesion was made in the normal eye during a monocular deprivation experiment, and found that the initially deprived eye would only learn visual information corresponding to the lesioned area. (Guillery, 1972). In particular, this experiment highlights that a multi-pathway deep linear network captures the competitive learning dynamics that have been empirically observed in many animal studies.

## 4 CRITICAL LEARNING PERIODS FOR MATRIX COMPLETION: GENERALIZATION IN DEEP LINEAR NETWORKS

In the previous section, we explored how competition between pathways affects how features are learned, finding that deficits early during training affect the learning of all features, while deficits later in training only affect features that have not been learned earlier in training. However, in the deep multi-pathway network, the network always learned the same global input-output mapping, even though the network learned to represent features differently across the different pathways depending on the onset of the deficit during training. The previous setup did not allow for a natural notion of generalization and an understanding of how generalization depends on the relationship between the data distribution during and after the deficit period.

To better understand generalization using minimal and tractable models, we turn to a matrix completion framework. Matrix completion is a general problem of imputing missing values in a matrix, given

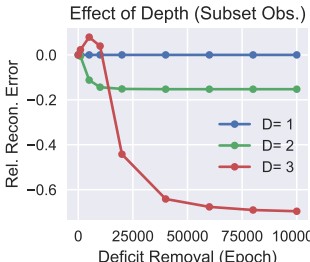 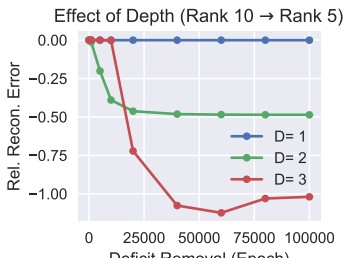

Figure 4: **Deeper networks have more pronounced critical periods. (Left)** Effect of variable depth parameterization on reconstruction error when observing partial observation of the final task (1000 out of $N$ entries) during the deficit period before the task switches and training continues by training on all $N$ observed entries. $N = 1500$. We computed the relative error by comparing the reconstruction error with a network that started training from the random initialization. A depth-1 parameterization is not affected (blue line), whereas deeper architectures are more affected by a sufficiently long initial deficit. Absolute reconstruction error was $0.864$, $0.320$ and $0.118$ for the depth 1, 2, 3 network trained from random initialization. **(Right)** Effect of variable depth parameterization when going from a rank 10 matrix completion task to a rank 5 matrix completion task. In this case, for a depth-1 parametrization, no effect is observed, whereas deeper networks are increasingly sensitive to perturbations early during training. Absolute reconstruction error was $0.795$, $0.053$ and $0.0001$ for the depth 1, 2, 3 network trained from random initialization.

some number of observed entries of the ground-truth matrix. The matrix completion setup, as we will see, is useful for studying critical learning period phenomena because it allows us to explicitly specify the relationship between tasks (during and after the deficit period) and allows flexibility for the type of deficit that can be applied. For matrix completion, given observed entries $\{M_{i,j} : (i,j) \in \Omega\}$ of unknown ground-truth matrix $M$, the challenge is to optimize a loss over a training set

$$L(W) = \frac{1}{2} \sum_{(i,j) \in \Omega} (M_{i,j} - W_{i,j})^2 \tag{9}$$

and generalize to the entries in unobserved locations. Typically the ground-truth matrix $M$ is assumed to be low rank to make the problem tractable.

Similar to the previous section, we parametrize the matrix $W$ using a deep linear neural network so what $W = W_D W_{D-1} \cdots W_1$, where $D$ refers to the depth of the parametrization and run gradient descent over this (over)parametrization. This setup has been used to study the implicit bias of SGD in a tractable setting (Gunasekar et al., 2017; Arora et al., 2019).

As we elaborate in Sec. 4.2 and Appendix B, we also obtain exact differential equations that characterize the evolution of the singular values (Eq. 10) and singular vectors during and after the deficit period (Eqs.13, 14) for matrix completion using a deep linear network parameterization.

## 4.1 EXPERIMENTAL DETAILS

In line with previous work (Arora et al., 2019), we initialize components by setting the standard deviation for each parameter in the deep matrix factorization to be $\sigma = \frac{1}{\sqrt{N}} \cdot g^{\frac{1}{D}}$, where $g$ sets the initial scale, and $N$ refers to the number of columns (or rows) of the square matrices $W_i$. This allows the Frobenius norm of the overall product matrix to be independent of the depth of the factorization. We consider ground-truth matrices of size $100 \times 100$, and matrices of the same size for all $\{W_i\}_{i=1}^{D}$ in the deep linear network parametrization. We set the number of observed entries to be 2000 and sample the same observations during the pre-training and final matrix completion task, unless otherwise stated for an experiment. We trained with SGD with constant learning rate of $0.2$ by using batch gradient descent and minimizing the loss in Eq. 9 averaged over observed entries.

Whereas previous work have studied matrix completion using SGD from small initializations (Gunasekar et al., 2017; Arora et al., 2019), we explore the case of transfer learning, where prior knowledge is embedded in the learned parameters. In particular we analyze how pre-training on one task (which we refer to as a deficit period) will affect generalization of a new task. We trained networks for a variable duration on the first task, and then subsequently trained the network for a

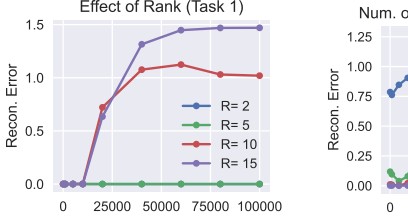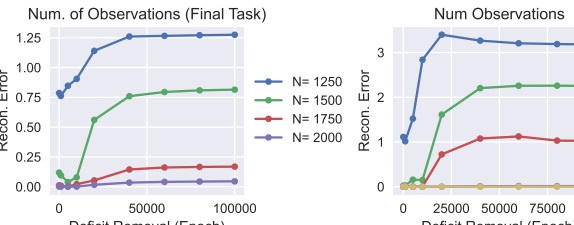

Figure 5: **Dependency on matrix rank and number of observed matrix entries. (Left)** In the matrix completion setup, sensitivity to an initial deficit occurs when the initial matrix completion task is higher rank than the final matrix completion task (rank 5). Pretraining on a rank 2 or rank 5 task does not adversely affect reconstruction error for the final matrix completion task (overlapping in the plot). **(Center)** Effect of partial observation of the final task (1000 out of $N$) during the deficit period before the task switches, and all $N$ observations as present. Generalization error as a function of $N$ in the final task. We observe a deficit when the number of samples in the final task are small, and "close" to the initial training task ($N = 1250$ and $N = 1500$). **(Right)** Sensitivity to initial training task as a function of total number of random observations $N$ for both tasks. The initial task is a rank 10 matrix completion task, and the final task is a rank 5 matrix completion task. The sensitivity to the change in task is most prevalent in low-sample regimes, where there are a variety of ways for fitting the observations, and is less impacted by the initialization coming from the initial learning. When the observations are large (N = 3000, or N = 4000), we do not observe sensitivity to the initial task (overlapping in plot).

fixed number of epochs of the final task (30000 additional epochs). On experiments where we vary the rank of the task during the initial training period, we set the rank of the first task to be 10, and the rank of the final task to be 5 by construction, unless otherwise stated.

## 4.2 RESULTS: IMPACT OF PRE-TRAINING ON GENERALIZATION FOR MATRIX COMPLETION

We first examine how the depth of the deep linear network parametrization in the matrix completion setup alters generalization on the final task, as a function of pre-training on an initial different task (Fig. 4). First, we find that deeper architectures are increasingly affected by training on a partial subset of the entries during an initial deficit period (Fig. 4, left). We also find that deeper networks are more affected by the deficit of pre-training on a different higher rank initial task (Fig. 4, right). We do not observe any sensitivity in depth $D = 1$ parametrization, as only components corresponding to observed entries get updated during both the pre-training and final task, and hence generalization to the unobserved entries will not be affected (being equivalently poor). Our finding that deeper architectures are more affected by an initially training on a related, corrupted dataset, is consistent with previous empirical results of Achille et al. (2019) and Kleinman et al. (2023) who empirically found that deeper convolutional architectures (with nonlinearities) were increasingly affected by initially training on blurred images before training on regular images.

Next we fixed the depth of the network to be $D = 3$, and varied the rank of the matrix during the pre-training task. We find critical learning periods when going from a higher rank task to a lower rank final task (Fig. 4, right), but not when going from a lower to higher rank tasks. We can better understand the learning dynamics by examining the singular values during the deficit periods and normal training. Arora et al. (2019) showed that starting from small initializations, the singular values will evolve as:

$$\dot{\sigma}_r(t) = -D \cdot \sigma_r(t)^{2-\frac{2}{D}} \cdot \mathbf{u}_r^T(t)\nabla(L(W(t))\mathbf{v}_r(t) \tag{10}$$

where $\mathbf{u}_r$ and $\mathbf{v}_r$ and the $r^{th}$ singular vector of the product matrix $W$ through training. In this manner, depth $D$ makes larger singular values increase faster, and makes smaller singular values evolve slower, and as in the previous section, the network will undergo sigmoidal learning trajectories for each singular value. In particular, we find that the ability to learn low-rank solutions relates to how well the network will generalize to unobserved entries (Fig. 6, top row), and deficits that lasted late into training prevents the learning of the low-rank solution for the final task. We also obtain closed form equations that describe how the singular vectors evolve during and after the deficit period (Eqs.13, 14), further described in Appendix B. Simulating these differential equations matches gradient descent learning dynamics before and after the task switches (Fig. 9).

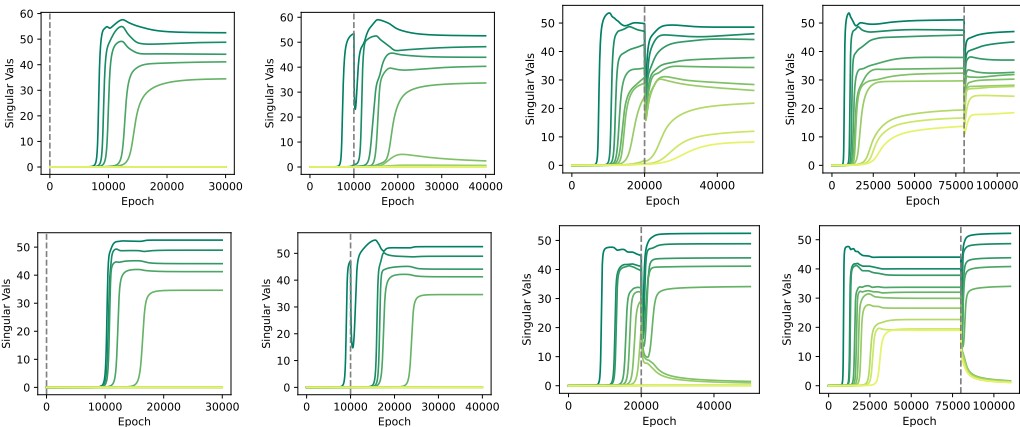

Figure 6: **Evolution of singular values before and after deficit as a function of the number of observed matrix entries.** The dashed gray lines indicate the epoch where the task changed from a rank 10 task to a rank 5 final task. **(Top row:** 2000 observations) If the deficit is applied early in training (up to epoch 10000) before the network has learned many singular modes, the network will converge to a low rank solution (and generalize well). Deficits late in training after many singular modes have been learned for the initial task lead to a solution that is not low rank and has worse generalization. **(Bottom row:** 4000 observations) With a large number of observations the network can eventually learn the correct final rank 5 task regardless of the initialization coming from training on the initial task. Regardless of the initialization, the network converges to a solution with exactly 5 singular values. Experiments are shown for a depth 3 network.

In the case of matrix completion, the sensitivity to the initial phase of training is most prevalent in low-sample regimes, as there are a variety of ways for fitting the observed entries. When the observations are large ($N = 3000$, or $N = 4000$), we do not observe sensitivity to the initial task (Fig. 5, right). With a large number of observations, the network can eventually learn the correct final rank 5 task, evidenced by the final singular values shown in Fig. 6, bottom. We also explore how the reconstruction error depends on the number of observations for the final task when initially observing a subset of the observed entries in Fig. 5 (center), and we observe critical learning periods in the low sample regime, where increasing duration of pre-training on partial task information increasingly impairs generalization performance for the final task.

## 5 DISCUSSION

The explanation most often ascribed to critical learning periods is that they emerge from factors unique to biology, such as biochemical processes that alter neural plasticity as animals age (Hensch, 2004). In this work, using minimal models that are analytically tractable, we argue that critical learning periods may be fundamental to *deep* learning systems and emerge as a result of information processing constraints, as they are even present in deep linear networks devoid of biochemical processes (and nonlinearities).

We find that multi-pathway deep linear networks capture the competitive dynamics between multisensory inputs, such as visual input coming into two eyes, and display maximum sensitivity to deficits to a pathway early during learning, as such defcits affect the learning of all features. Further, the effect of such deficits are increasingly pronounced in deeper networks. We also find that generalization in the matrix completion setting is increasingly altered by longer pre-training on a different task, and this effect too depends critically on the depth of the over-parametrization. Together, our analysis highlights that critical periods depend on two main factors: the depth of the model and the structure of the data distribution, which allows us to make a strong correspondence with biological systems that share these details.

Our models are minimal and analytically tractable, and yet surprisingly are still sufficient to recapitulate much of the critical period phenomena seen in biological systems. Our work thus makes an important step towards developing a mathematical understanding of critical learning periods as a fundamental phenomenon common to artificial and biological learning systems that depends primarily on the data distribution and the depth of the network learning to process such information.

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

## A    ADDITIONAL EXPERIMENTAL DETAILS

### A.1    MULTIPATHWAY EXPERIMENTS

In the multi-pathway experiments, when we applied the deficit to a pathway, which we refer to as "blocking" or "gating" an input to a pathway in the paper, the desired target output was also shifted by a baseline amount corresponding the deprived pathway's output. This was to ensure that the normal pathway was only required to learn the unexplained component of the output, and not the entire output. In practice we implemented this using the `.detach()` method in PyTorch applied on the output of the deprived pathway during the deficit period, while having the total output be the sum of the contribution of both pathways (while leaving the desired target unchanged). In particular, this deficit corresponds to a deficit where the gradient was not supplied to the deprived pathway during the deficit window.

The task we consider in Fig. 2 and Fig. 3 is the following, with $\mathbf{\Sigma}^x = \mathbf{I}$. Each input is 8 dimensional and the output is 15 dimensional, with the input encoded as a one-hot vector, and the output corresponding to the columns of $\mathbf{\Sigma}^{yx}$ (rows of $\mathbf{\Sigma}^{yx\,T}$).

$$
\mathbf{\Sigma}^{yx} = \begin{pmatrix}
1 & 1 & 0 & 1 & 0 & 0 & 0 & 1 & 0 & 0 & 0 & 0 & 0 & 0 & 0 \\
1 & 1 & 0 & 1 & 0 & 0 & 0 & 0 & 1 & 0 & 0 & 0 & 0 & 0 & 0 \\
1 & 1 & 0 & 0 & 1 & 0 & 0 & 0 & 0 & 1 & 0 & 0 & 0 & 0 & 0 \\
1 & 1 & 0 & 0 & 1 & 0 & 0 & 0 & 0 & 0 & 1 & 0 & 0 & 0 & 0 \\
1 & 0 & 1 & 0 & 0 & 1 & 0 & 0 & 0 & 0 & 0 & 1 & 0 & 0 & 0 \\
1 & 0 & 1 & 0 & 0 & 1 & 0 & 0 & 0 & 0 & 0 & 0 & 1 & 0 & 0 \\
1 & 0 & 1 & 0 & 0 & 0 & 1 & 0 & 0 & 0 & 0 & 0 & 0 & 1 & 0 \\
1 & 0 & 1 & 0 & 0 & 0 & 1 & 0 & 0 & 0 & 0 & 0 & 0 & 0 & 1
\end{pmatrix}^T \tag{11}
$$

ADDITIONAL DETAILS FOR FIGURE 2.

We used a fixed length deficit of 150 epochs starting at epochs $\{0, 400, 750\}$ (early, middle, and late deficit respectively). We use batch gradient descent, with a learning rate of 0.01 and squared error loss. We did not use biases in the linear networks. We are able to exactly trace the trajectories of the singular values for the task in Fig. 2 provided the following initialization from (Saxe et al., 2013). We initialize weights matrices such that $\mathbf{W}_a^1 = \mathbf{R}\mathbf{D}\mathbf{V}^T$, $\mathbf{W}_a^i = \mathbf{R}\mathbf{D}\mathbf{R}^T$ and $\mathbf{W}_a^{D_a} = \mathbf{U}\mathbf{D}\mathbf{R}^T$ (and analogously for pathway B). We set $\mathbf{R}$ to be a $100 \times 8$ orthogonal matrix, $\mathbf{D}_a$ to be a diagonal matrix, and $\mathbf{U}$, $\mathbf{V}^T$ to be the singular vectors of $\mathbf{\Sigma}^{yx}$. The scale of the entries in diagonal matrices $\mathbf{D}$ at initialization was set to $0.01^{1/D_a}$ and we used the same diagonal matrix in both pathways. We added a small amount of Gaussian noise to the diagonal entries so that equivalent singular values were learned at slightly different times. This initialization ensures that the weights are balanced across layers, and the differential equations then characterizes the learning dynamics before, during, and after the deficit period.

ADDITIONAL DETAILS FOR FIGURE 3.

We used a fixed length deficit of 200 epochs starting at epoch 0 for the gated pathway. We also lesioned the second singular mode from the otherwise normal pathway by setting the second row and column of $K_a$ to zero during training. We also use batch gradient descent, with a learning rate of 0.01 and squared error loss. We did not use biases in the linear networks. We used a multi-pathway network with $D_a = D_b = 3$. The scale of the entries in diagonal matrices $\mathbf{D}$ at initialization was set to $0.01^{1/D_a}$ and we used the same diagonal matrix in both pathways. We added a small amount of Gaussian noise to the diagonal entries so that equivalent singular values were learned at slightly different times. We initialize weights matrices such that $\mathbf{W}_a^1 = \mathbf{R}\mathbf{D}\mathbf{V}^T$, $\mathbf{W}_a^i = \mathbf{R}\mathbf{D}\mathbf{R}^T$ and $\mathbf{W}_a^{D_a} = \mathbf{U}\mathbf{D}\mathbf{R}^T$ (and analogously for pathway B). We set $\mathbf{R}$ to be a $100 \times 8$ orthogonal matrix.

### A.2    MATRIX COMPLETION EXPERIMENTS

We constructed an $N \times N$ matrix $M$ of particular rank $R$ by creating two $R \times N$ matrix $L$ and $L'$ each with entries sampled from a zero-mean Gaussian distribution with standard deviation 1, and then

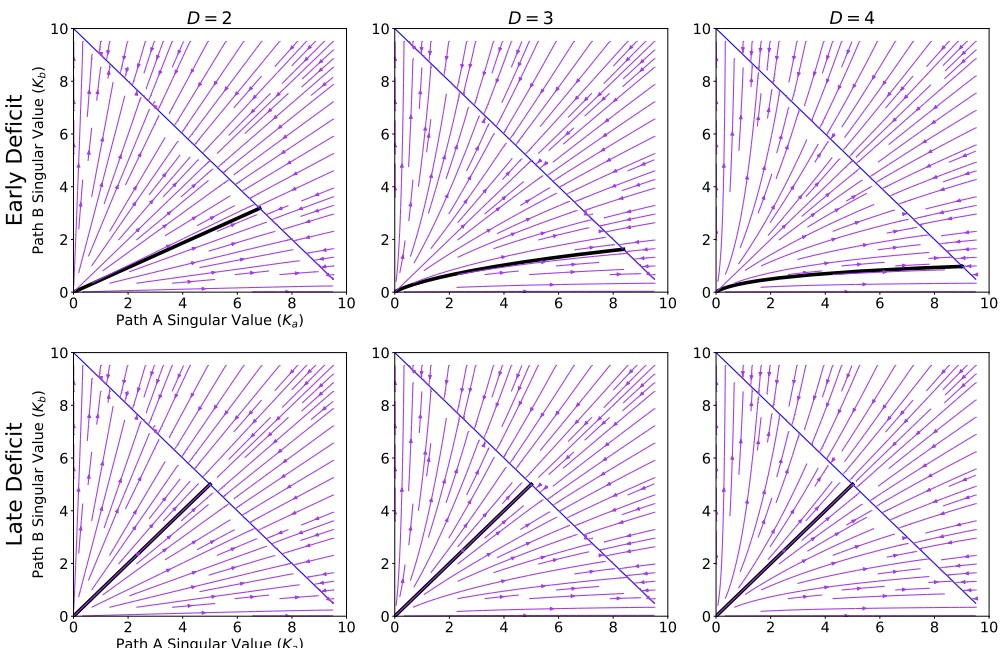

Figure 7: **Deeper networks are more affected by a temporary early deficit, even for small initial conditions.** Phase portrait of pathway specific singular value $K_{a\alpha}$ and $K_{b\alpha}$ where $K_{a\alpha} + K_{b\alpha} = \sigma_\alpha$, where $\sigma_\alpha = 10$ is the singular value and denoted by the diagonal blue line in the plots. Black traces indicate a simulation for a particular initial condition where $q(0) = p(0) = \epsilon$, and $\epsilon = 0.005$. Flow fields (purple arrows) are shown for systems without a deficit. **(Top row)** An early deficit to pathway B (for only 5 epochs) leads to the other pathway learning more of the feature, and this effect becomes increasingly more pronounced in deeper networks (shown for depths ranging from 2 to 4). **(Bottom row)** In contrast, late deficits (even of 10000 epochs, but starting at epoch 500000) have a negligible effect on the final solution, where neither pathway dominates (on average) how a feature is learned. Step size of 0.01 was used and simulation was run for $T = 1000000$ epochs.

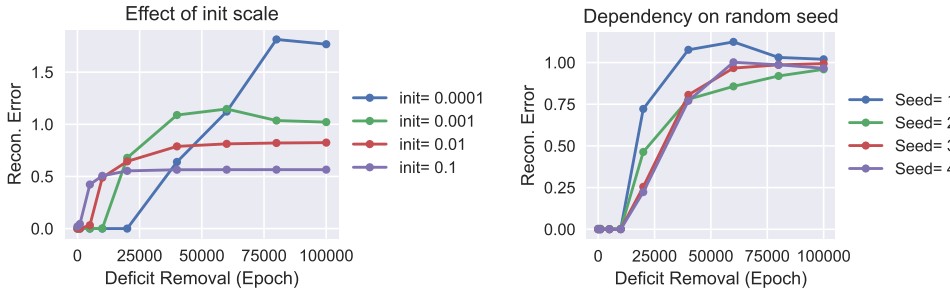

Figure 8: **Effect of initialization scale and robustness to random initialization. (Left)** We used the same parameter settings as in the paper, and varied the scale of the initialization. For all initializations, we see higher reconstruction with longer pre-training on the initial task. Smaller initializations have a longer period (number of epochs) where the deficit will not impact the final solution, but have a more marked final effect (e.g. blue trace). **(Right)** Results are robust to different random initialization seed (for initialization scale of 0.001).

taking $\tilde{M} = L'^T L$. This matrix was normalized such that $M = \frac{\tilde{M}}{||\tilde{M}||_F} \cdot \frac{N}{R}$, where $|| \cdot ||_F$ denotes the Frobenius norm. In Fig. 6 we plot the top 10 singular values computed from the product matrix $W = W_D W_{D-1} \cdots W_1$ during training. In our experiments, rather than optimize the sum of all the losses as in Eq. 9, we computed the loss as the average. Reconstruction error was measured by $\frac{1}{N^2}||M - W||_F^2$ (average squared error per entry). We used an initial scale $g = 0.01$ in our experiments unless otherwise stated, where entries were drawn from a normal distribution with standard deviation $\sigma = \frac{1}{\sqrt{N}} \cdot g^{\frac{1}{D}}$. We vary this initialization in Fig. 8. All other experimental details to reproduce experiments are provided in main text.

### A.2.1 ADDITIONAL EXPERIMENTS: INITIALIZATION SCALE AND ROBUSTNESS TO RANDOM INITIALIZATION

We vary the scale of the initialization in Fig. 8 (left), and observe critical learning periods across different initialization shapes. Notably, smaller initializations have a longer period where an initial deficit will not impact the final solution, but have a more marked final effect with a long deficit. In these experiments, once the task switched, we continued training for $50,000$ epochs, which was sufficient training duration for the small initializations to converge without a deficit.

We also find that our learning dynamics are robust to different initializations (Fig. 8, right).

### A.3 COMPUTE TIME

All experiments in the paper can be reproduced on a local computer in around 7 hours. We used a 2017 Macbook Pro (3.1 GHz Quad-Core Intel Core i7).

## B EXACT DIFFERENTIAL EQUATIONS CHARACTERIZE MATRIX COMPLETION LEARNING DYNAMICS BEFORE AND AFTER TASK TRANSFER

We consider the singular value decomposition of the product matrix $\mathbf{W}(t) = \mathbf{W}_D(t)\mathbf{W}_{D-1}(t) \cdots \mathbf{W}_1(t) = \mathbf{U}(t)\mathbf{A}(t)\mathbf{V}^\top(t)$. The following equations describe how the effective singular values $\mathbf{A}(t)$ and the singular vectors $\mathbf{U}(t)$ and $\mathbf{V}(t)$ evolve over time, under certain conditions (Arora et al., 2019), which we will elaborate on:

$$\tau \dot{a}_r(t) = -D \cdot a_r(t)^{2-\frac{2}{D}} \cdot \mathbf{u}_r^T(t)\nabla(L(\mathbf{W}(t))\mathbf{v}_r(t) \tag{12}$$

$$\tau \dot{U}(t) = -U(t)\left(F(t) \odot \left[U^\top(t)\nabla\ell(W(t))V(t)A(t) + A(t)V^\top(t)\nabla\ell^\top(W(t))U(t)\right]\right)$$
$$- \left(I - U(t)U^\top(t)\right)\nabla\ell(W(t))V(t)(A^2(t))^{\frac{1}{2}-\frac{1}{D}} \tag{13}$$

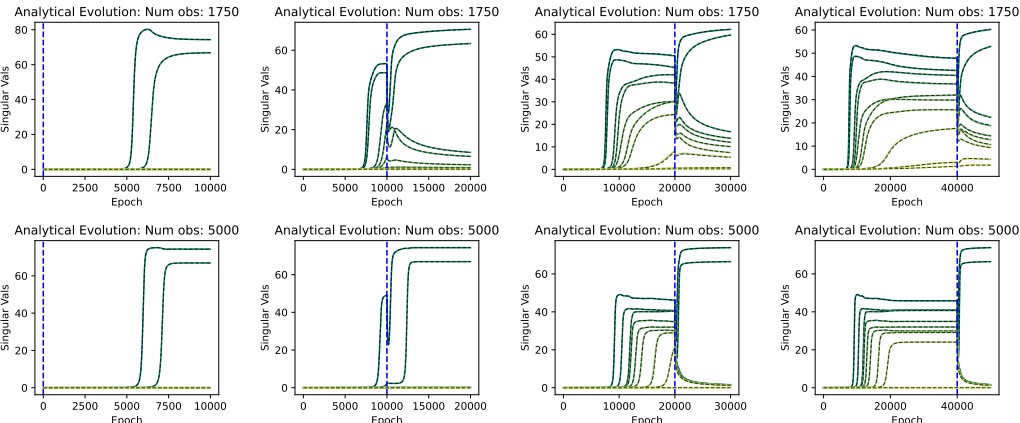

Figure 9: **Correspondence between learning dynamics through differential equations and neural network training.** Simulating dynamics from the differential equations from Eq. 12,13, 14 (dashed lines) matches simulations obtained with gradient descent training (colored lines). The blue dashed lines indicates the epoch where the task switches. **(Top Row:)** Given a small number of observations (1750), the network converges towards a low rank solution when the deficit is removed early, while if the deficit is removed late, the network retains previously learned information (singular modes). **(Bottom row:)** Given a large number of observations (5000), the network converges to a low-rank solution regardless of when the deficit is removed. For these simulations, the initial ground truth matrix $M_1$ was a rank 8 matrix, and the final target matrix was a rank 2 matrix $M_2$. After changing tasks (indicated by vertical blue dashed line), we continue training for 10000 epochs. Additional parameters used: $D = 3$, $\lambda = 0.25$, $M_1, M_2 \in \mathbb{R}^{N \times N}$ with $N = 100$.

$$
\begin{aligned}
\tau \dot{V}(t) \quad = \quad & -V(t)\left(F(t) \odot \left[A(t)U^{\top}(t)\nabla\ell(W(t))V(t) + V^{\top}(t)\nabla\ell^{\top}(W(t))U(t)A(t)\right]\right) \\
& - \left(I - V(t)V^{\top}(t)\right)\nabla\ell^{\top}(W(t))U^{\top}(t)(A^2(t))^{\frac{1}{2}-\frac{1}{D}} ,
\end{aligned}
\tag{14}
$$

where $\odot$ indicates element-wise product, $D$ indicates the depth of the network, $F(t)$ is a skew-symmetric matrix with $((\sigma_{r'}^2(t))^{1/N} - (\sigma_r^2(t))^{1/N})^{-1}$ in entry $(r, r')$ where $r \neq r'$, and $I$ refers to an identity matrix.

These equations describe the evolution of the weights starting from a balanced initialization $(W_{j+1}^T(0)W_{j+1}(0) = W_j(0)W_j^T(0) \,\forall\, j)$, and assuming that singular values are distinct and different from zero (Arora et al., 2019). The balanced initialization is approximately satisfied when starting from small initial conditions (Arora et al., 2019).

The gradient of loss can be computed in closed form for matrix completion with squared error loss.

$$
\nabla(L(\mathbf{W}(t)) = \begin{cases} -(M_{ij} - W_{ij}) & \text{if } (i, j) \in \Omega, \\ 0 & \text{otherwise.} \end{cases}
\tag{15}
$$

Importantly, the conditions hold even when the task switches, as in our critical learning period experiments. This is because the difference of $\mathbf{W}_{j+1}^T(t)\mathbf{W}_{j+1}(t) - \mathbf{W}_j(t)\mathbf{W}_j^T(t)$ is a constant of training for all $j$ (Arora et al., 2018). We show in Fig. 9, the simulations obtained through the differential equations match the simulations obtained by training a neural network SGD, both for the periods before and after changing tasks. (In practice we initialized each matrix as a diagonal matrix with distinct and small values to ensure the conditions were satisfied.)

## C  COMPARISON AGAINST CLOSED-FORM LEARNING DYNAMICS FOR COMPLETE MATRIX OBSERVATIONS

Let product matrix $\mathbf{W}(t) = \mathbf{U}(t)\mathbf{A}(t)\mathbf{V}^T(t)$. (Arora et al., 2019, Theorem 3) showed that its singular values evolve as:

$$
\tau \dot{a}_r(t) = -D \cdot a_r(t)^{2-\frac{2}{D}} \cdot \mathbf{u}_r^T(t)\nabla(L(\mathbf{W}(t))\mathbf{v}_r(t)
\tag{16}
$$

Note that we added a time constant $\tau$.

Using the loss $L(\mathbf{W}) = \frac{1}{2} \sum_{(i,j) \in \Omega} (\mathbf{M}_{i,j} - \mathbf{W}_{i,j})^2$ from matrix completion, where $\mathbf{M}$ refers to the ground-truth matrix, the gradient is:

$$\nabla L(\mathbf{W}_{ij}(t)) = -[\mathbf{M} - \mathbf{W}(t)]_{i,j} \tag{17}$$

in the observed entries $(i, j) \in \Omega$ and 0 otherwise in unobserved locations.

Consider the case when all entries are observed and $\mathbf{M} = \mathbf{U}\mathbf{S}\mathbf{V}^T$ and $\mathbf{W}(t) = \mathbf{U}\mathbf{A}(t)\mathbf{V}^T$, then:

$$\nabla L(\mathbf{W}(t)) = -\mathbf{U}(\mathbf{S} - \mathbf{A}(t))\mathbf{V}^T \tag{18}$$

and substituting Eq. 18 into the first equation Eq. 10, we will get:

$$\tau \dot{a}_r(t) = D \cdot a_r(t)^{2 - \frac{2}{D}} \cdot (s_r - a_r(t)) \tag{19}$$

This corresponds to the differential equation from (Saxe et al., 2013, Eq. 15). For a depth $D = 2$ network (Saxe et al., 2013) found that the effective singular value of the product matrix

$$\mathbf{W}(t) = \mathbf{W}^2(t)\mathbf{W}^1(t) = \mathbf{U}\mathbf{A}(t)\mathbf{V}^T = \sum_\alpha a_\alpha(t)\mathbf{u}^\alpha \mathbf{v}^{\alpha T} \tag{20}$$

would evolve in the following manner where

$$a_\alpha(t) = \frac{s_\alpha e^{2s_\alpha t/\tau}}{e^{2s_\alpha t/\tau} - 1 + s_\alpha/a_\alpha^0}. \tag{21}$$

This equation precisely specifies the dynamics if the weights are initialized to lie in the SVD basis where $\mathbf{W}(t = 0) = \epsilon \mathbf{U}\mathbf{V}^T$.

### C.1 EXTENSION TO TRANSFER BETWEEN TASKS.

Consider a setting where we first are given a subset of entries from matrix $\mathbf{M}_A$ and we train first the first epochs, and then switch or transfer to a different matrix $\mathbf{M}_B$ which we continue training on for subsequent epochs. This setting corresponds to one where have two ground truth matrices $\mathbf{M}_A$ and $\mathbf{M}_B$, corresponding to two tasks $A$ and $B$. For the first task, task $A$, training will proceed as described in the previous section.

If the singular vectors are shared between tasks, $\mathbf{M}_A = \mathbf{U}\mathbf{S}_A\mathbf{V}^T$ and $\mathbf{M}_B = \mathbf{U}\mathbf{S}_B\mathbf{V}^T$ (but the ordering changes e.g. corresponding singular value), we can use the exact same dynamics from above to solve for transfer between tasks.

### C.2 EXPERIMENTS

In Fig. 10 we show the predictions from Eq. 21, and the results obtained from simulating a neural network in the case of a full observations and partial observations. In particular, we simulate a neural network of depth $D = 2$ for a matrix completion task of matrix dimension $N \times N$, where $N = 100$. For the first 10000 epochs, the ground truth matrix was a rank 6 matrix $\mathbf{M}_a$, and for the subsequent epochs, the desired task was a rank 5 matrix $\mathbf{M}_b$. The matrices are constructed so that $\mathbf{M}_a = \mathbf{M}_b + s_r \mathbf{u}_r \mathbf{v}_r^T$, where $s_r = 20$. We trained networks with a learning rate $\lambda = 0.5$, initialization scale $\epsilon = \sqrt{a_0} = 0.01$, and time constant $\tau = \frac{N^2}{\lambda}$, where the $N^2$ factor is because we used the average matrix completion loss in our implementation (as opposed to the sum in Eq. 9). The closed-form solution from Eq. 21 matches the simulation when all matrix entries are observed, and also approximately matches simulations when the number of entries is relatively large (Fig. 10, middle). When the number of entries is small, the network learns a different solution (Fig. 10, right).

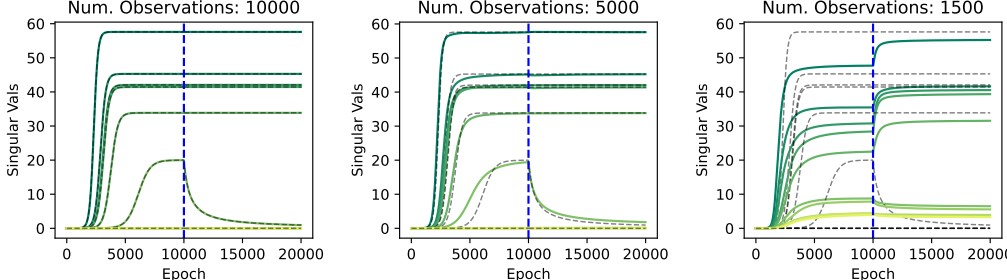

Figure 10: **Comparison against closed-form learning dynamics for complete matrix observations.** When all entries of matrix are observed, we can obtain an exact closed-form solution for the evolution of singular values (shown in gray dashed lines). The simulations of training a neural network are shown in colored lines. For the first 10000 epochs, the ground truth matrix was a rank 6 matrix $\mathbf{M}_a$, and for the subsequent epochs, the desired task was a rank 5 matrix $\mathbf{M}_b$. The change in task is denoted by the blue dashed lines. The matrices are constructed so that $\mathbf{M}_a = \mathbf{M}_b + s_r \mathbf{u}_r \mathbf{v}_r^T$, where $s_r = 20$. **(Left)** If all entries are observed, the analytical prediction matches simulation and we do not observe a critical period, as observed in Fig. 6. **(Middle)** Even with a fraction of observations (5000 entries which is 50% of the total entries), the analytical predictions assuming all entries are observed for both tasks closely match simulation. **(Right)**. Given a small number of entries (1500) the network does not eventually learn a minimum rank solution, and as shown in Fig. 5 and Fig. 6, is associated with worse generalization and critical learning periods.

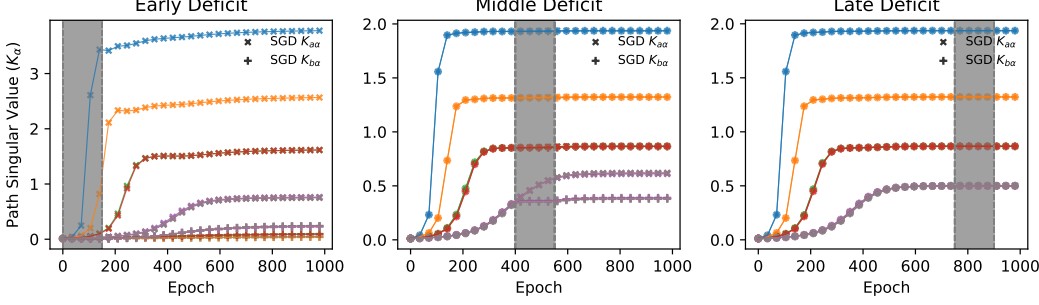

Figure 11: **Early deficits affect learned representations in multi-pathway model using the Tanh nonlinearity, while late deficits do not.** We consider a network with $D_a = D_b = 3$, with otherwise the same settings as in Fig. 2. **(Left)** Early gating deficit (denoted by gray period; epoch 0 to 150) to pathway B leads to all features being learned in pathway A. When training with SGD, we indicate the singular values for pathway with crosses ('X'), and pathway B with a plus sign ('+'), with the different colors identifying different singular values. We observe sigmoidal learning trajectories of the singular values, with larger singular values learned earlier in training. **(Middle)** Deficits applied in the middle of training only affects previously unlearned features (bottom four singular modes), whereas other singular values are learned equally in both pathways (crosses and plus signs overlapping for blue, orange, red, green singular modes). **(Right)** Late deficits (epoch 750 to 900) has a negligible effect on how features are learned, and results in features being learned equally in both pathways (crosses and plus signs overlapping).

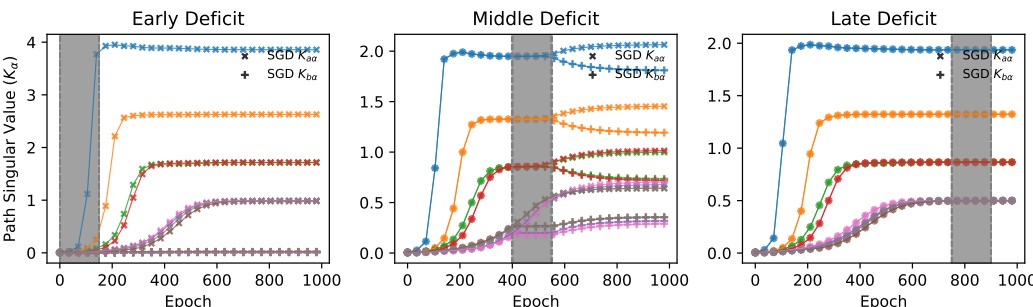

Figure 12: **Early deficits affect learned representations in multi-pathway model with the Relu nonlinearity, while late deficits do not.** We consider a network with $D_a = D_b = 3$, with otherwise the same settings as in Fig. 2. **(Left)** Early gating deficit (denoted by gray period; epoch 0 to 150) to pathway B leads to all features being learned in pathway A. When training with SGD, we indicate the singular values for pathway with crosses ('X'), and pathway B with a plus sign ('+'), with the different colors identifying different singular values. We observe sigmoidal learning trajectories of the singular values, with larger singular values learned earlier in training. **(Middle)** Deficits applied in the middle of training affects previously unlearned features (bottom four singular modes). Features that are partially learned (blue, orange, red, green singular modes) are slightly affected, in contrast to the setting without nonlinearity (Fig. 2). **(Right)** Late deficits (epoch 750 to 900) has a negligible effect on how features are learned, and results in features being learned equally in both pathways (crosses and plus signs overlapping).

