# OpenReview forum: "Critical Learning Periods Emerge Even in Deep Linear Networks"
_ICLR.cc/2024/Conference — ICLR 2024 spotlight_

### Official Review · Reviewer_qaj6 · 2023-10-30

**Soundness:** 3 good
**Presentation:** 3 good
**Contribution:** 3 good
**Rating:** 8
**Confidence:** 3

**Summary:**

This paper reveals that critical learning periods are not limited to biological learners
but are fundamental to the learning itself. Through well-written experiments, the
authors show how deep linear networks display critical learning periods. They
conducted two experiments: one focused on competition in a multi-path model
(similar to our eyes), and the other focused on the effect of critical periods on transfer
learning.

The first experiment reveals that a deprivation deficit in one pathway will result in the
other pathway to winning the competition and learning all the corresponding features.
The deprivation only has an effect if applied in the early epochs. This reinforces the
critical learning period claim. The authors also show, by permanently lesioning the
dominant pathway, that the deprived pathway isn’t dead and will pick up features that
the dominant pathway has missed.

The second experiment highlights the effect of depth and data distribution on the
impact that deprivation during the critical period has. It reveals that deeper networks
are more sensitive to deprivations in the critical period, and the structure of the data
distribution also dictates the impact of the deprivation.

**Strengths:**

1. The paper provides a mathematical understanding of the psychological concept of the “critical learning period”.
2. The paper reveals that critical learning is not restricted to biological learners but is inherent to learning itself!
3. Theoretically strong.
4. Well-thought-out and well-delivered experiments.
5. The paper neatly highlights the pros and cons of competition between a model’s branches.
6. By introducing a permanent lesion in the dominant branch, the authors show that the deprived branch isn’t dead for the rest of the training.
7. The authors draw convincing insights from the transfer learning experiments, both low-to-high rank and high-to-low rank. In the appendix, the authors have also covered (both theoretically and experimentally) the case where two different matrices are involved.

**Weaknesses:**

> …our analysis shows that critical periods only depend on two main factors: the depth of the model and the structure of the data
distribution, as opposed to details of the architecture and optimization problem.
1. The authors prove the dependence on depth and data distribution. But how can
they claim independence from other factors, like architecture and non-linearities,
especially when considering models other than deep linear networks?

2. Figure 2’s markers should be changed. It is difficult to differentiate between a
circle and a circle-triangle overlap. Try plus (+) and cross (x) as markers or a
straight line (|) and an oblique line (/).

3. Codebase not provided.

4. Typos:
* Page 3: had minimum the → had the minimum
* Page 6: training has only affects → training only affects

------------------------------

Overall, I am appreciative of the work done. I would like the authors to release the codebase for reproducibility, and clarify some of the concerns raised. I would be happy to revisit the rating if these concerns are addressed.

**Questions:**

> In the multi-pathway experiments, when we applied the deficit to
a pathway, which we refer to as “blocking” or “gating” an input to
a pathway in the paper, the desired target output was also shifted
by a baseline amount corresponding the deprived pathway’s
output. This was to ensure that the normal pathway was only
required to learn the unexplained component of the output, and
not the entire output.

I understood the use of PyTorch’s “.detch()” for blocking the flow of gradients in the
deprived branch. However, shifting the desired target output to ensure learning the
unexplained component did not make sense to me. Please elaborate.

---

> ### Author Response · Authors · 2023-11-19
> **Response to Reviewer qaj6**
>
> We thank the reviewer for the detailed comments and accurate summary of our work. We also thank the reviewer for the concrete presentation suggestions. We describe how we address their comments below.
>
> > **The authors prove the dependence on depth and data distribution. But how can they claim independence from other factors, like architecture and non-linearities, especially when considering models other than deep linear networks?**
>
> The reviewer is correct that the point is misphrased. Indeed, critical periods can depend, in part, on other factors (such as optimization algorithm and non-linearities), which has also been shown empirically in nonlinear networks (Achille et al., 2019). We rephrased the point in our introduction and discussion. We want to emphasize, however, that even ablating those details (nonlinearities and optimization algorithm), as done in our paper, we still observe critical periods that depend on architecture depth and data distribution. This means that the effect of other factors is not necessary for the emergence of critical periods in deep networks, and in general may have a less significant effect than architecture depth and data distribution.
>
> In our revised paper, we ran additional experiments in our multipathway setup where we included nonlinear activation functions (Relu and Tanh) and observed similar trends to in the deep multipathway linear case (Fig 11 and Fig 12 in revision).
>
> >**Figure 2’s markers should be changed. It is difficult to differentiate between a circle and a circle-triangle overlap. Try plus (+) and cross (x) as markers or a straight line (|) and an oblique line (/).**
>
> We thank the reviewer for the suggestion. We modified our multipathway plots accordingly, and we believe it helps with distinguishing between the different pathways.
>
> > **Codebase not provided.**
>
> We will provide our code along with the final version of our paper.
>
> >**Typos:**
>
> We fixed the typos. Thanks for the careful reading and for catching those!
>
> >**I understood the use of PyTorch’s “.detch()” for blocking the flow of gradients in the deprived branch. However, shifting the desired target output to ensure learning the unexplained component did not make sense to me. Please elaborate.**
>
> Detaching is how it is implemented in our code. The alternative (and equivalent) interpretation as shifting the target was provided solely to relate it to a more biologically meaningful process. Specifically, consider a deficit where parameters along pathway A are not updated. Without the deficit the Loss is $L_{normal} = (y - W_A^{tot}x - W_B^{tot}x)^2$, and gradients are updated in both pathways. During the deficit (to pathway A) the loss is equivalent to $L_{def} = (\tilde{y} - W_B^{tot} x )^2$ where $\tilde{y} = y - W_A^{tot}x$, and gradients only occur in pathway B. We referred to $\tilde{y}$ as the shifted output.
>
> ---
> Achille, Alessandro, Matteo Rovere, and Stefano Soatto. "Critical learning periods in deep networks." International Conference on Learning Representations. 2019.

---

> > ### Comment · Reviewer_qaj6 · 2023-11-19
> > **thanks for the clarifications but concerned on reproducibility and transparency**
> >
> > I appreciate the clarifications and modifications made by the authors to address the concerns raised. However, I am concerned that the authors have not released the codebase despite this being one of the specific requests being made. I believe reproducibility of ML works is important for transparency and trust. This prohibits me from raising the rating further.

---

> > > ### Author Response · Authors · 2023-11-19
> > > **We have now provided anonymized code**
> > >
> > > Thank you for your constructive feedback. We are particularly appreciative of your responsiveness and consideration in revising your score after reading our response. We have anonymously uploaded our code for our multipathway experiments here: https://anonymous.4open.science/r/CriticalPeriodLinearMultiPath-AFBB and for the matrix completion experiments here: https://anonymous.4open.science/r/AnonCriticalPeriodMatrixCompletion-2AF7

---

> > > > ### Comment · Reviewer_qaj6 · 2023-11-19
> > > > **thanks for sharing the code**
> > > >
> > > > I am happy with the revision and all my concerns are now addressed. I will increase my rating to accept.

---

### Official Review · Reviewer_uHcm · 2023-11-01

**Soundness:** 3 good
**Presentation:** 2 fair
**Contribution:** 2 fair
**Rating:** 6
**Confidence:** 3

**Summary:**

This paper demonstrates that critical learning periods in deep learning models depend on the depth of the model and the structure of the data distribution, offering an analytical perspective on the learning dynamics. This study provides both analytical and simulation evidence that the learning of features is closely tied to the competition between different sources of information, highlighting the competitive dynamics within the learning process.

**Strengths:**

1. This paper is well-written, technically solid and has a concrete flow.
2. This work offers insights into the reasons behind the emergence of critical learning periods in both biological and artificial networks.

**Weaknesses:**

1. I think the assumption of the deep linear network is a bit strong. Since in real-world applications, most neural networks require non-linear activations. A deep linear network can be actually approximated by a single-layer linear network.
2. Critical periods in artificial deep neural networks (DNNs) may be due to specificities of the optimization process, such as an annealing learning rate, or from defects in the artificial implementation and training, like ReLU units becoming frozen or gradients vanishing.
3. What's the model's generalization ability on more sparse data settings?

**Questions:**

Please consider the things listed in the “Weaknesses” section.
Also please consider providing information regarding the limitation and future work of this paper.

---

> ### Author Response · Authors · 2023-11-19
> **Response to Reviewer uHcm**
>
> We thank the reviewer for their time, feedback, accurate summary of our work, and comments. We address their concerns below.
>
> >**I think the assumption of the deep linear network is a bit strong. Since in real-world applications, most neural networks require non-linear activations. A deep linear network can be actually approximated by a single-layer linear network.**
>
> While it is true that a deep linear network has the same expressive power as a single-layer linear network, we emphasize that the parameterization of a deep linear network leads to complex and depth-dependent non-linear learning dynamics that is sufficient to lead to the emergence of critical periods. By restricting our analyses to deep linear networks, we highlight that the deep parameterization alone (which is shared with non-linear networks) is enough to explain the emergence of critical periods and may indeed be the most important factor underlying them. Indeed, in our updated experiments in our revision paper that we posted, we show that even a non-linear network follows largely the exact same trends as the multipathway linear network in our experiments (Figure 11, Figure 12).
>
> >**Critical periods in artificial deep neural networks (DNNs) may be due to specificities of the optimization process, such as an annealing learning rate, or from defects in the artificial implementation and training, like ReLU units becoming frozen or gradients vanishing.**
>
> We agree and acknowledge this directly in our introduction as one of the main motivations for studying deep linear networks in our paper. In particular we show that deep linear networks trained with constant learning rate and without nonlinearities also exhibit analogous phenomena, and also permit analytical analysis. As we replied to R1, we don't want to claim that in any model depth and structure of the data distribution are the only factors determining the existence of critical periods. In fact, other papers have shown that choice of optimization algorithm or regularization can significantly affect the critical period (Achille et al., 2019). What we show is that depth and structure of the data are a minimal set of requirements that alone can support the emergence of a critical period, and also enable differential equations that precisely characterize the learning dynamics. We clarified the text to better communicate this.
>
> >**What's the model's generalization ability on more sparse data settings?**
>
> We interpret this question to be referring to our matrix completion settings. In Fig. 5 right, we examine reconstruction error as a function of the number of entries. In general matrix completion becomes easier with a larger number of observed entries. When we have a very sparse number of entries (1000 entries) the network obtains a large reconstruction error even without pre-training on a different task (this corresponds to the point deficit removal at 0). In the sparse setting too, the network obtains worse reconstruction by initially training on the higher rank task (other points along the curves).
> We also found that pre-training on a subset of the observed entries adversely affects generalization (Fig 5, left). We added in an additional experiment to highlight that this effect also depends on the depth of the parameterization, with deeper architectures being more affected (Fig 13).
>
> ---
> Achille, Alessandro, Matteo Rovere, and Stefano Soatto. "Critical learning periods in deep networks." International Conference on Learning Representations. 2019.

---

> ### Comment · Reviewer_uHcm · 2023-11-22
>
> I thank the author for the detailed response, which has addressed some of my concerns. I have increased the score accordingly.

---

### Official Review · Reviewer_PVne · 2023-11-08

**Soundness:** 4 excellent
**Presentation:** 4 excellent
**Contribution:** 3 good
**Rating:** 10
**Confidence:** 4

**Summary:**

Noting the anecdotal and empirical evidence of critical periods in both biological and artificial learning agents, the authors hypothesize that critical learning periods are a general feature of learning, and not an accident of biology or training, They investigate this hypothesis analytically via deep linear networks, which retain many characteristics of typical neural networks while being amenable to tractable analysis. They perform a variety of analyses and experiments, largely inspired by work on critical periods in biological systems, demonstrating that depth and data distribution provide information processing constraints that give rise to the critical periods.

**Strengths:**

Well written, interesting avenue of research. Results well explained early in the text, with clear examples and clear explanations of experimental settings and technical details. Particularly nice clarity given the cross-disciplinary nature of the work.
Related work is well written, reasonably thorough (to my knowledge), and well-explained.
Nice phase portraits.
Captions are mostly well-explained and self contained! (I don't know why this is so rare, but good job)

**Weaknesses:**

It's a bit of a tradeoff with having a clear hypothesis, but I find the case oversimplified/misframed  in the beginning and conclusion. Sure the literal biochemical explanation doesn't hold in an artificial setting, but analogies of this (reduced plasticity) could well occur. I don't think these are 'competing' hypotheses as they are framed; plasticity, imperfect optimization, etc. are *mechanisms* by which critical periods could arise in both artificial and biological systems; it doesn't tell us the reason  (inherent to learning, quirks of substrate, something else, etc.). In the conclusion "natural information processing constraints" are contrasted with biochemical processes like plasticity as an alternative hypothesis, but what is (decreased) plasticity buyt a natural information processing constraint? Demonstrating presence in artificial systems doesn't estabilsh it as a general feature of learning, except insofar as we've defined learning narrowly to be done by biological and ANNs (does naive bayes learn? does it exhibit critical periods?). It becomes sort of a circular argument about what does "learning" even mean. This commonality is actually supported by your experiments in Fig 3 recreating Guillery's. Anyway, I think the key sentences/claims (e.g. last of the abs) and results stand and are extremely interesting,  I would just like to see this framed around mechanisms or instantiations of information processing constraints, or something, rather than competing hypotheses. and maybe acknowledge more of the ambiguity of what "learning systems in general" means/ (without compromising clarity and narrative, which I believe are strong points of the paper).

Terms used in equations are not explained when introduced  in 3.1 -- I know it's from a citation and you might be short on space, but you will lose a lot of readers here and it's a shame not to continue the clear prose you've had thus far.

Around eqn 9, connection of matric completion to what you're trying to do is not clear -- be explicit about how imputing missing values is the same as / allows us to specify  task relationships with flexibility.

While the relationship of experiments to showing dependence on depth is very clear and well-explained, the insights and relationship of experiments to data distribution is much less clear.

**Questions:**

See "weaknesses" for some less concrete suggestions (feel free to tell me what you plan to write if you address them and I'm happy to provide feedback!)

The main other thing I'd like to see is connections made/discussed with nonlinear networks -- what should we expect to change vs. hold? maybe according to the sources you cite (e.g. Saxe), or ideally, a suite of experiments with e.g. MLPs of increasing depth.
Another specific link could be made in Fig. 6 to scaling laws.

small things:
 - first sentence shouldnt repeat abstract
 - 'competition between pathways affect' -> affects
- inconsistent use of "artificial neural network", "deep net", "neural net", et al. For me it's fine, but could be confusing for someone outside the field; better to be consistent
- title for plot should be removed (conflicts/causes potential confusion vs caption and X axis label, nonstandard for technical works to have an embedded title)
 - remind us what "Path Singular Value" is in the caption/change caption e.g .singular value dimensions (confusing that its referred to as 1-5 but labelled 0-4 on plot and called something different). All this stuff explaining the axis and basics of the plots should go first, before the interpretation. - I didn't understand the comment "and do not affect performance on the final gask" in fig $. Does it mean they have equal performance?

---

> ### Author Response · Authors · 2023-11-19
> **Response to Reviewer PVne (1/2)**
>
> We thank the reviewer for the accurate summary of our work, and their time, feedback, and insightful comments. We appreciate the positive assessment of our research and presentation of our results. We explain below how we address the reviewer’s comments.
>
> >**Framing of introduction and conclusion:**
>
> We appreciate this thoughtful and insightful point. We updated the introduction and discussion to better highlight this nuance – indeed “natural information processing constraints” should not be contrasted with biochemical processes. In our introduction and discussion, we aim to emphasize that architecture and data distribution are sufficient for leading to critical periods in deep artificial networks, and that this offers an intriguing, and different lens for understanding critical periods in biological systems. We plan to further elaborate on this in subsequent revisions subject to space constraints and incorporating other reviewer feedback during the discussion period.
>
> >**Terms used in equations are not explained when introduced  in 3.1 -- I know it's from a citation and you might be short on space, but you will lose a lot of readers here and it's a shame not to continue the clear prose you've had thus far.**
>
> We updated our manuscript to better introduce notation (also pointed out by other reviewers).
>
> >**Around eqn 9, connection of matric completion to what you're trying to do is not clear -- be explicit about how imputing missing values is the same as / allows us to specify  task relationships with flexibility.**
>
> Thanks, we updated our manuscript to better introduce the matrix completion setting – the changes are in blue text of our revision.
>
> >**While the relationship of experiments to showing dependence on depth is very clear and well-explained, the insights and relationship of experiments to data distribution is much less clear.**
>
> We appreciate the feedback. In general, pre-training on a modified data distribution for some duration can either be helpful or harmful, depending on the relationships between tasks. In the case of critical learning periods, we focus on the case when such pre-training is harmful. Empirically, Achille et al., (2019) found that pre-training on blurred images was harmful for subsequent generalization on normal images, but pre-training on pure noise for the same duration was much less harmful. This suggests that the learning dynamics depend on the complex relationship between the tasks.
>
> In general it’s difficult to define the “complexity of a task” or the relationships between tasks. However, the matrix completion setting gives us the ability to completely specify the task, its complexity, and the relationship between tasks. We find that pre-training on a higher rank task (more complex) leads to worse generalization performance (Fig 5, left). We also found that pre-training on a subset of the observed entries (a partial view of the data, similar to the blurred image experiments) also leads to impaired generalization (Fig 6, left). We added in an experiment to highlight that this effect becomes more pronounced in deeper architectures (Fig 13 of revision).
>
> We updated the text to better motivate these experiments in text.
>
> >**The main other thing I'd like to see is connections made/discussed with nonlinear networks -- what should we expect to change vs. hold? maybe according to the sources you cite (e.g. Saxe), or ideally, a suite of experiments with e.g. MLPs of increasing depth. Another specific link could be made in Fig. 6 to scaling laws.**
>
> We expect deeper networks to be more affected by a temporary deficit, and we now directly discuss this connection in the text. This finding that deeper architectures are more affected by an initially training on a related, partial dataset, is consistent with the empirical results of Achille et al., (2019, in particular see Fig 2, right panel of reference) and Kleinman et al., 2023 (in particular see Fig 5 center panel of reference) who empirically found that deeper convolutional architectures were increasingly affected by initially networks on blurred images before training on regular images. We also ran an additional experiment where we trained on a subset of the observed entries, and we found that while a single layer network is not affected, deeper architectures are increasingly affected by a sufficiently long initial deficit (Figure 13 of revision) We thank the reviewer for the helpful suggestion.
>
> > **inconsistent use of "artificial neural network", "deep net", "neural net"**
>
> We thank the reviewer for the feedback and will better clarify the text. There were times where we wanted to emphasize the distinction between artificial and biological networks, as well as highlight the depths of the architectures, but we will strive for improved consistency and clarity.

---

> ### Author Response · Authors · 2023-11-19
> **Response to Reviewer PVne (2/2)**
>
> >**remind us what "Path Singular Value" is in the caption/change caption e.g .singular value dimensions (confusing that its referred to as 1-5 but labelled 0-4 on plot and called something different).**
>
> Path Singular Value refers to the contribution of each pathway to the singular value and corresponds to to $K_a$ (or $K_b$) defined in Sect 3.1. We now better describe this in the text. $K_a + K_b$ sum to the singular values at convergence. In Fig. 3, we only plot the top 5 task singular values – we clarified the text.
>
> >**I didn't understand the comment "and do not affect performance on the final task" in fig [4]. Does it mean they have equal performance?**
>
> We clarified the sentence in our revision. Indeed, it means they lead to equivalent performance on the final task.
>
> ---
>
> Achille, Alessandro, Matteo Rovere, and Stefano Soatto. "Critical learning periods in deep networks." International Conference on Learning Representations. 2019.
>
> Kleinman, Michael, Alessandro Achille, and Stefano Soatto. "Critical Learning Periods for Multisensory Integration in Deep Networks." Proceedings of the IEEE/CVF Conference on Computer Vision and Pattern Recognition. 2023.

---

### Official Review · Reviewer_SR9U · 2023-11-10

**Soundness:** 3 good
**Presentation:** 3 good
**Contribution:** 3 good
**Rating:** 5
**Confidence:** 3

**Summary:**

This paper is a continuation of the study of critical learning periods in deep learning. This work focuses on why critical learning periods emerge in deep linear network models. The paper investigates the critical learning period that depends on the depth of the model and structure of the data distribution, with supporting experiments. Meanwhile, they also show the learning of features is tied to competition between sources. They analyze the impact of pre-training on some tasks, and there are some simple experiments used to prove the “critical learning periods”. This work provides analytical understanding of critical periods in deep linear networks and draws connections between artificial and biological learning.

**Strengths:**

1. This paper continues previous work (on the deep network), the experiments correspond to research on the depth of the network, data distribution, competition between sources, and pre-training.
2. This work on studying the competition of different data sources is solid, and it seems to be a relatively good job through the linear multi-pathway network.
3. Analytical and minimal models provide fundamental insight. Intuition and empirical observations match well.

**Weaknesses:**

1. The paper is less readable and requires a higher theoretical foundation. The description of the “linear multi-pathway framework” (Sec.3.1.) is not clear enough and lacks corresponding details.
2. Only studying deep linear networks seems not enough to clearly understand why deep networks have critical periods, and network depth and data sources cannot guarantee sufficient persuasiveness.
3. There are fewer categories of experiments, and it would be better if the authors could provide more types of experiments to prove their claims.
4. For the experiments related to pre-training, I cannot guarantee sufficient correlation with key learning practices. I hope I can explain the motivation for doing this part of the experiment. By the way, this is not to say that this is an obvious weakness.
5. I don’t find any support in the paper for what the abstract said: “Why critical learning periods emerge in biological and artificial networks”.

**Questions:**

Please see the Weaknesses.

---

> ### Author Response · Authors · 2023-11-19
> **Response to Reviewer SR9U (1/2)**
>
> We thank the reviewer for their accurate summary of our work and for their time, feedback, and comments. We explain below how we address the reviewer’s comments.
>
> >**Improved clarity of the ``linear multi-pathway framework" (Sec.3.1.)**
>
> In our revised manuscript, we better introduce our notation in Sect 3.1. The updates are highlighted in blue text. We thank the reviewer for pointing this out, and we believe the changes lead to a more readable manuscript.
>
> >**Only studying deep linear networks seems not enough to clearly understand why deep networks have critical periods, and network depth and data sources cannot guarantee sufficient persuasiveness.**
>
> We don't want to claim that in any model depth and structure of the data distribution are the *only* factors determining the existence of critical periods. In fact, other papers have shown that choice of optimization algorithm or regularization can affect the critical period (Achille et al., 2019, Zilly et al., 2021). What we show instead is that depth and structure of the data are a minimal set of requirements that alone can support the emergence of a critical period, and also enable differential equations that precisely characterize the learning dynamics. This allowed us to better understand how depth, competition between sources, and data distribution affects critical learning periods. We have clarified the text in our revision.
>
> >**There are fewer categories of experiments, and it would be better if the authors could provide more types of experiments to prove their claims.**
>
> We ran additional experiments to further support our claims that we include in our revision. Related to the above question, for the multipathway simulations, we included results using Relu and Tanh nonlinearity and observed similar conclusions where deficits early on as in the deep linear case (Figure 11, and Figure 12 in revision)
>
> For the matrix completion setup, we also ran an additional experiment where we trained on a subset of the observed entries, and we found that while a single layer network is not affected, deeper architectures are increasingly affected by a sufficiently long initial deficit (Figure 13). This finding that deeper architectures are more affected by an initially training on a related, partial dataset, is consistent with previous empirical results of Achille et al (2019; in particular see Fig 2, right panel of reference) and Kleinman et al (2023;  in particular see Fig 5, center panel of reference) who empirically found that deeper convolutional architectures (with nonlinearities) were increasingly affected by initially training on blurred images before training on regular images.
>
> >**For the experiments related to pre-training, I cannot guarantee sufficient correlation with key learning practices. I hope I can explain the motivation for doing this part of the experiment. By the way, this is not to say that this is an obvious weakness.**
>
> In neuroscience, critical periods are observed in two settings broadly speaking: (1) sensory deprivation, when the input to one sensor is completely absent, and (2) the related setting when the agent initially learn on a abnormal data distribution (e.g., because of anomalous working of the sensor, or abnormal sensory experience) before having access to proper data later in life. While our multi-pathway experiments are targeted at modeling the competition-based effects coming from the first case, our matrix completion setup where we pre-train on a modified data distribution setting more closely models the second case (the network is pretrained on a different data distribution than the one it sees later in training). The matrix completion setup provides a good simple model to probe this critical period phenomena (how generalization on a final task is affected by pre-training on an initial task) since it allows a notion of generalization, as well as a well defined notion of tasks, their complexity, and their relationships.
>
> **Please continue to the comment below**

---

> > ### Author Response · Authors · 2023-11-19
> > **Response to Reviewer SR9U (2/2)**
> >
> > >**I don’t find any support in the paper for what the abstract said: “Why critical learning periods emerge in biological and artificial networks”.**
> >
> > Previous work emphasized several factors in the emergence of critical periods in deep networks such as the optimization algorithm and the non-linearities (Achille et al, 2019). However, these factors are absent from biological systems, thus making it difficult to establish a direct connection. We show that depth of the system is one of the main contributing factors to critical periods in DNNs, and is a more fundamental aspect which helps connecting with biological systems. This observation that deeper architectures are more prone to critical learning periods is consistent with neuroscience experiments beginning with Hubel and Wiesel that find that low-level sensory tasks (such as vision), which have deep circuits processing such information, are prone to critical periods.
> >
> > ---
> > Achille, Alessandro, Matteo Rovere, and Stefano Soatto. "Critical learning periods in deep networks." International Conference on Learning Representations. 2019.
> >
> > Zilly, Julian, et al. "On plasticity, invariance, and mutually frozen weights in sequential task learning." Advances in Neural Information Processing Systems 34 (2021): 12386-12399.
> >
> > Kleinman, Michael, Alessandro Achille, and Stefano Soatto. "Critical Learning Periods for Multisensory Integration in Deep Networks." Proceedings of the IEEE/CVF Conference on Computer Vision and Pattern Recognition. 2023.

---

### Author Response · Authors · 2023-11-19
**General Response**

We thank all the reviewers for their thoughtful comments and feedback. We believe their feedback has led to an improved manuscript, now posted. Changes in our revision are denoted with blue text. In particular, we include additional experiments for both the multipathway setup and matrix completion setting, to better highlight the connection of our work with deep networks with nonlinearities.

For the multipathway simulations, we included results using Relu and Tanh nonlinearity and observed similar conclusions as in the deep linear case (Figure 11, and Figure 12 in revision). For the matrix completion setup, we also ran an additional experiment where we trained on a subset of the observed entries, and we found that while a single layer network is not affected, deeper architectures are increasingly affected by a sufficiently long initial deficit (Figure 13). This finding that deeper architectures are more affected by an initially training on a related, partial dataset, is consistent with previous empirical results of Achille et al (2019) and Kleinman et al (2023) who empirically found that *deeper* convolutional architectures (with nonlinearities) were increasingly affected by initially training on blurred images before training on regular images.

We also fixed minor typos and made presentation improvements in response to the specific reviewer comments.

---
Achille, Alessandro, Matteo Rovere, and Stefano Soatto. "Critical learning periods in deep networks." International Conference on Learning Representations. 2019.

Kleinman, Michael, Alessandro Achille, and Stefano Soatto. "Critical Learning Periods for Multisensory Integration in Deep Networks." Proceedings of the IEEE/CVF Conference on Computer Vision and Pattern Recognition. 2023.

---

### Meta-Review · Area_Chair_gRxj · 2023-12-08

**Metareview:**

The reviewers unanimously acknowledge that the paper contributes significantly to the understanding of critical learning periods in deep linear network models. They appreciate the analytical approach taken to investigate the emergence of these critical periods, particularly how model depth and data distribution constraints play a pivotal role. The experiments conducted provide sufficient evidence supporting the existence of critical learning periods, with a focus on competitive dynamics within the learning process. Additionally, the experiments highlight the importance of early epochs for deprivation effects and demonstrate the impact of network depth and data distribution on critical periods. Overall, the paper is commended for bridging the gap between artificial and biological learning, shedding light on the fundamental nature of critical learning periods.

There were some concerns related to the recent state-of-the-art non-linear neural networks, but the contribution and scope of this work are mostly on leveraging the deep linear structure to draw the connection with biological learning systems. This work provides an intriguing and novel perspective for understanding critical periods in biological systems.

**Justification For Why Not Higher Score:**

This work provides a very interesting perspective to draw the connection between biological and artificial networks.
Despite the interesting contribution of this work, the reviewers were expected to extend a more practical setup, e.g., the connection to non-linear networks. Although the authors indeed did a good job of trying to expand their results to the non-linear ones during the rebuttal period, those are empirical and preliminary and the original development is limited to the linear one.

**Justification For Why Not Lower Score:**

This work provides an intriguing and novel perspective for understanding critical periods in biological systems.

---

### Decision · Program_Chairs · 2024-01-16

Accept (spotlight)